# Changes in and Predictors of HIV among People Who Inject Drugs in Mizoram, Northeast India, from 2007 to 2021

**DOI:** 10.3390/ijerph20105871

**Published:** 2023-05-19

**Authors:** Lucy Ngaihbanglovi Pachuau, Caterina Tannous, Richard Lalramhluna Chawngthu, Kingsley Emwinyore Agho

**Affiliations:** 1School of Health Science, Western Sydney University, Campbelltown Campus, Campbelltown, NSW 2560, Australia; 2Mizoram State AIDS Control Society, Mizoram 796012, India; 3Translational Health Research Institute (THRI), Western Sydney University, Campbelltown Campus, Penrith, NSW 2571, Australia; 4Faculty of Health Sciences, University of Johannesburg, Doornfontein Campus, Johannesburg 2094, South Africa

**Keywords:** human immunodeficiency virus, HIV, HIV infection, India, people who inject drugs, injecting drug users

## Abstract

This study aimed to examine the changes in and predictors of the human immunodeficiency virus (HIV) among people who inject drugs (PWID) in Mizoram, Northeast India, over a period of 15 years (2007–2021). A sample of 14783 PWID was extracted from the Targeted Intervention (TI) services under the Mizoram State AIDS Control Society (MSACS). A chi-square test was used to compare the differences in HIV prevalence across the three 5-year periods, and a multiple logistic regression analysis was used to determine predictors after adjusting for sociodemographic, injecting and sexual behaviours. The results showed that compared to 2007–2011, HIV prevalence was almost three times higher in 2012–2016 (AOR 2.35; 95% CI 2.07–2.66) and almost two times higher in 2017–2021 (AOR 1.41; 95% CI 1.24–1.59). The results suggest that participants who were females (AOR 2.35; 95% CI 2.07–2.66), married (AOR 1.13; 95% CI 1.00–1.27), separated/divorced/widowed (AOR 1.74; 95% CI 1.54–1.96), of middle school level education (AOR 1.24; 95% CI 1.06–1.44), sharing needles/syringes (AOR 1.78; 95% CI 1.61–1.98) and receiving a regular monthly income were positively associated with HIV infection. Condom use with a regular partner (AOR 0.77; 95% CI 0.70–0.85) was high among PWID. Despite targeted interventions under MSACS to reduce HIV in Mizoram, the prevalence of HIV/AIDS among PWID remained high between 2007 and 2021. Policymakers and stakeholders should tailor future interventions based on the factors identified in this study that are associated with HIV infection. Our findings highlight the importance of socio-cultural factors in HIV epidemiology among PWID in Mizoram.

## 1. Introduction

Human immunodeficiency virus (HIV) in India was first detected in 1986 among female sex workers in Chennai [1]. The epidemic in India continued to grow and peaked in the late nineties when India was found to have the third-largest HIV epidemic in the world [2]. HIV sentinel surveillance (HSS) conducted annually across India, monitors the trends, levels and burden of HIV among different population groups and helps to inform effective responses to control HIV/AIDS [3]. HIV prevalence in India is high among those who have unprotected sexual contact with multiple sexual partners and people who inject drugs (PWID) [3]. Female sex workers (FSWs), men who have sex with men (MSM), transgender people (TG), long-distance truck drivers and migrants are also high-risk groups.

In 2020, of an estimated 2.3 million people living with HIV (PLHIV) in India, 0.22% were adults aged 15–49 years. Since the peak 1997 period, overall HIV infections have declined by 89% [4]. New HIV infections in India declined by 33.3% between 2010 and 2020. Between 2010 and 2019, AIDS-related deaths declined in India by almost 66% compared to the global average of 39% [5].

The Care, Support and Treatment (CST) programme implemented under the National AIDS Control Programme (NACP) in 1992 has been at the core of India’s successful AIDS response [5]. Among PWID, the focus on opiate substitution treatment (OST), the syringe needle exchange programme (SNEP), the increased availability of condoms and the treatment of STIs have been important components of HIV prevention programmes [6]. The NACP provision of HIV testing services at more than 31,000 facilities across India has assisted with the early detection of HIV infections [7]. This programme also offers anti-retroviral treatment (ART), free of charge for people diagnosed with HIV [7]. The fall in the number of AIDS-related deaths is largely due to the increase in ART coverage [8].

Despite the decrease in overall HIV prevalence in India, numbers continue to be high (more than 1%) in the three northeastern states. In 2020, Mizoram was estimated to have the highest HIV prevalence (2.37%) followed by Nagaland (1.44%) and Manipur (1.15%) [4]. Among the different high-risk groups (HRGs), injecting drug users are the main concern in these northeastern states [9]. Many of the northeastern states in India share an easily penetrable border with Myanmar, which facilitates the trafficking of heroin into Mizoram, Manipur and Nagaland [9,10]. According to the national report on substance use in India, there were 28,288 PWID in Mizoram in 2019 [11]. From the various aspects of demography and unsafe injecting behaviour, Mizoram appears to be the state with the highest risk of HIV transmission [9].

Targeted Interventions (TIs), under the previously described NACP framework, were first introduced and implemented in Mizoram in 2007. These included behaviour change communication, treatment of sexually transmitted infections, distribution of condoms and other risk reduction materials, needle exchange programs, opioid substitution therapy, ownership building and creating an enabling environment [12,13].

However, despite these interventions, Mizoram still reported the highest incidence of HIV/AIDS in India in 2020 [4]. An investigation into the factors that may contribute to these high numbers in Mizoram has not yet been undertaken, thus supporting the need for this study. This study is not an evaluation of intervention programmes. More specifically, this study aims to examine the changes in and predictors of HIV infection among PWID in Mizoram over a 15-year period. Findings from this study will guide policymakers and service providers in the re-design of programmes and interventions to reduce the rate of new HIV infections.

## 2. Materials and Methods

### 2.1. Study Design and Setting

This study used a cross-sectional design, analysing secondary data from participants registered in the Targeted Intervention (TI) programme under the Mizoram State AIDS Control Society (MSACS). Datasets from January 2007 through to February 2021 were used to calculate the number of PWID diagnosed with HIV infection in Mizoram. The year 2007 was chosen as the baseline as this was the year that TIs were first implemented in Mizoram. A total of 14,783 PWID participants were registered in the TI programme between 2007 and 2021.

Secondary data were collected from 34 TI-NGOs in eight districts (8) in Mizoram. Recruitment of participants in the TI programme involved mapping exercises to identify hotspot areas where PWID congregated in different districts of Mizoram. Outreach workers (ORWs) and peer educators (PEs) visited hotspot areas to invite PWID to enrol in TI services. After several encounters, PEs and ORWs invited PWID to enrol in the TI services. Once consent was given, each newly recruited participant was given a unique personal identification number and was registered and enrolled in the TI programme [14]. Pre- and post-test counselling and HIV testing were performed by a nurse and a laboratory technician in an Integrated Counselling and Testing Centre (ICTC) or in a Community Base Screen (CBS) escorted by a PE or ORW [15]. Individuals who reported injecting drugs three months prior to the date of data collection were eligible participants for enrolment into TI services.

### 2.2. Study Population

The study population consisted of all PWID aged 18 years and above registered in a TI programme under MSACS and newly diagnosed with HIV between January 2007 and February 2021 in Mizoram.

### 2.3. Study Region

The region for this study is Mizoram. Mizoram is a small state in northeast India, with Aizawl as its capital. The state of Mizoram shares a 722-kilometre border with the neighbouring countries of Bangladesh and Myanmar. It has a population of 1,091,014. It is the second least populous state in India and covers an area of approximately 21,087 square kilometres [16].

### 2.4. Ethical Considerations

This study obtained ethics clearance and approval (No.D.12019/1/2020-MSACS (RA)) from the MSACS.

### 2.5. Outcome and Exploratory Variables

The outcome of interest in the study was HIV infection among PWID and was coded as binary 1 for ‘Yes’ and 0 for ‘No’. The exploratory variables selected for this study were influenced by previous studies on HIV prevalence among PWID [17,18,19] and were classified into three main factors, namely, sociodemographic factors, injecting behaviour and sexual behaviour. The sociodemographic characteristics included age in category (‘18–24’, ‘25–34’ and 35+), gender (male/female), marital status (never married, married, separated/divorced/widowed), educational status (primary, middle, higher, graduate and above), employment status (unemployed, employed, self-employed) and average monthly income in Indian rupees (INR) (none, <3000, 3001–6000, 6001–10,000, >10,000). Injecting behaviour factors included sharing of needle/syringe (Yes/No). Factors related to sexual behaviour were whether the person used a condom with a regular partner (Yes/No).

## 3. Statistical Analysis

Categorical data were summarized as counts and percentages for all sociodemographic characteristics and injecting and sexual behaviours for every 5-year period in the MSACS data, followed by determining the trends in the prevalence of HIV among PWID in Mizoram over a 15-year period. We divided the 15 years into 5 years for three time periods (2007–2011, 2012–2016 and 2017–2021) because monitoring and impact evaluation of population-based health surveys in many low- to middle-income countries are conducted every 3–6 years [20]. In India, the Demographic and Health Surveys (DHS) conduct national surveys every 5 years to allow comparisons over time; hence, we followed the standard DHS survey style [20,21]. The statistical method used by Agho et al. (2016) [22] was used to examine differences in prevalence. The three time periods were classified into three categories. Category 1 arbitrarily referred to the 2007–2011 time period, category 2 to the 2012–2016 time period and category 3 to the 2017–2021 time period. Furthermore, to determine the changes between categories ‘1 and 2’, category 1 was coded as ‘1’ while category 2 was coded as ‘0’, and a similar procedure was carried out to determine the changes between categories ‘1 and 3’ and categories ‘2 and 3’. In our analysis, we created the survey weight to be equal to 1, and the survey mean command in STATA was used to compare each time period and exploratory variables, as reported in Table 1, and to determine the prevalence estimates. Differences in prevalence estimates of HIV among PWID were expressed as percentages comparing the data across the three 5-year periods. To determine the comparisons between the three 5-year periods and report the significant differences, the linear combinations of parameters (lincom) command in STATA was used to determine the significance of differences at *p* < 0.05 for each of the sociodemographic characteristics and injecting and sexual behaviours, as reported in Table 2.

Bivariate analyses were performed to examine the independent association between the outcome and exploratory variables (sociodemographic characteristics and injecting and sexual behaviours). Multiple logistic regression models were fitted to examine the factors associated with HIV among PWID in Mizoram. In the univariate analysis, all exploratory variables with a *p*-value < 0.20 were retained and used to build multiple logistic regression models. For multiple logistic regression, a manual elimination process was used to remove non-significant variables (*p* > 0.05). Only those variables with *p* < 0.05 were regarded as factors associated with HIV among PWID. The odds ratio (OR) and 95% confidence intervals were obtained from the adjusted logistic regression models and were used to measure factors associated with HIV among PWID in Mizoram. All analyses were performed using commands in STATA version 17.0 (Stata Corporation, College Station, TX, USA).

## 4. Results

### 4.1. Sociodemographic Characteristics and Injecting and Sexual Behaviours of Study Participants (Years 2007–2021)

The sociodemographic characteristics and injecting and sexual behaviours of PWID for the three time periods 2007–2011, 2012–2016 and 2017–2021 are shown in Table 1. Between 2007 and 2021, a total of 14,783 PWID were registered in a TI program under the MSACS. Of these, the proportion of males did not change significantly, remaining well above three-quarters of the population. The proportion of females increased from 4.84% in 2007–2011 to 10.59% in 2012–2016. Throughout all three time periods, the proportion of PWID increased for those aged 25–34 years and 35 years or older, who had 10–12 years of education, and who were self-employed. There was also an increase in the average monthly income between the income bracket of INR 6001–10,000 and above INR 10,000. The proportion of PWID who shared needles increased from 2017 to 2021 (44.49%). There was a gradual decrease in the proportion of PWID who used condoms with a regular partner from 2007 to 2021.

### 4.2. Changes in the Proportion of HIV (2007–2021)

The overall changes in the proportion of HIV among PWID for the periods 2007–2011, 2012–2016 and 2017–2021 are shown in Figure 1. Figure 1 shows the overall decline in predicted HIV proportion among PWID from 2007 to 2011. However, from the year 2012, the proportion of PWID with HIV continued to increase significantly every year to the year 2016. There was a sharp drop in the proportion of HIV among PWID in 2017, and this continued to decline each year until 2021.

### 4.3. Changes in HIV Prevalence among PWID by Sociodemographic Characteristics and Injecting and Sexual Behaviour (2007–2021)

The change in HIV prevalence among PWID in sociodemographic characteristics and injecting and sexual behaviour between the three time periods 2007–2011, 2012–2016 and 2017–2021 is shown in Table 3. Overall, there was a significant increase in HIV prevalence in both male and female PWID (7.7%, *p* < 0.001 and 2.34%, *p* < 0.001, respectively). HIV prevalence increased significantly among PWID in all categories of age groups, but the increase was most significant among those 35 years and older (16%, *p* < 0.001). HIV prevalence increased among PWID who were married (9.7%, *p* < 0.001) and those who were separated/divorced/widowed (16.7%, *p* < 0.001). The prevalence of HIV increased among PWID who had primary (15.3%), middle (9.4%) and higher (7.1%) levels of education (*p* < 0.001 for all three education levels). HIV prevalence increased significantly among employed (11.2%, *p* < 0.001) and self-employed (13.4%, *p* < 0.001) PWID. The income level between INR <3000 and INR <10,000 increased significantly among HIV-positive PWID, but the increase was highest in the INR 6001–10,000 income bracket (12.9%, *p* < 0.001). The result showed a significant increase in sharing of needles among HIV-positive PWID (2.3%, *p* < 0.001). Condom use with a regular partner decreased (−1.6%, *p* < 0.001) among HIV-positive PWID.

### 4.4. Multivariable Analysis

Table 3 shows the unadjusted and adjusted odds ratios for the factors associated with HIV among PWID from 2007 to 2021. Only the factors identified as significant were included in the multivariable analysis. Overall, the prevalence of HIV was highest between the years 2012 and 2016 (AOR = 2.35, 95% CI 2.07–2.66). HIV infection remained positively associated with being female (AOR = 2.35, 95% CI 1.81–2.44). Being married (AOR = 1.13, 95% CI 1.00–1.27) and being divorced/separated/widowed were associated with increased odds of HIV infection. PWID with middle school level education (7–9 years) (AOR = 1.24, 95% CI 1.06–1.44) had higher odds of HIV infection. Employed participants had a significant association with higher odds of HIV infection (AOR = 1.14, 95% CI 1.03–1.27). Average monthly income had a positive association with HIV infection for all income categories including INR < 3000 (AOR = 2.14, 95% CI 1.91–2.42), INR 3001–6000 (AOR = 1.42, 95% CI 1.23–1.64), INR 6001–10,000 (AOR = 1.56, 95% CI 1.29–1.89), INR > 10,000 (AOR = 1.43, 95% CI 1.11–1.83). Sharing needles/syringes remained positively associated with HIV infection (AOR = 1.78, 95% CI 1.61–1.98). The participants who used condoms with a regular partner had lower odds of HIV infection (AOR = 0.77, 95% CI 0.70–0.85).

## 5. Discussion

This study was one of the first to use the MSACS datasets to examine changes in the prevalence and predictors of HIV among PWID between 2007 and 2021. In this study, it was observed that the proportion of HIV among PWID was three times higher in 2012–2016 and two times higher in 2017–2021 compared to the 2007–2011 time period. Similar patterns were observed for the adjusted odds ratios, which, compared to the 2007–2011 time period, were 2.75 and 1.88 times more likely to report HIV among PWID in the 2012–2016 and the 2017–2021 time periods, respectively. We observed a significant decrease in the HIV proportion among PWID during 2007–2011, followed by a steep increase from 2012 to 2016 and a steady decrease from 2017 to 2021. This study also found that being female, being married and divorced/separated/widowed, having a monthly income, being employed and sharing needles/syringes were associated with significantly higher odds of HIV among PWID in Mizoram.

In this study, we observed that the proportions of HIV among PWID were higher between 2012 and 2016 than between 2007 and 2011 and between 2017 and 2021. The reason for the increase could be multifactorial. Firstly, this increase reported in our study could be attributed to the increased availability of heroin at a low price, making heroin more affordable for new users in Mizoram. A study on trends in HIV/AIDS incidence and mortality between 1990 and 2017 conducted in Iran [23] reported an increased rate of narcotic drug injections due to the availability of low-priced heroin from Afghanistan and Pakistan, the major global producers of heroin. According to a United Nations report from 2000, geographical proximity to the ‘golden triangle’ of heroin production (Myanmar, Thailand and Laos), coupled with a very permeable Indo–Myanmar border security led to an easy passageway for drug smuggling, and this may have fuelled much higher rates of drug use in Mizoram [24]. Further research is needed to establish this finding, and such research should look at drug use patterns, program funding, harm reduction services and socio-structural changes during this period in Mizoram.

Second, opioid substitution therapy (OST) as an HIV prevention strategy among PWID [25] was rolled out free of charge in Mizoram in 2009; however, it was initially implemented and supported only in three non-governmental organization (NGO) centres and targeted only 360 PWID. The scale-up expansion of OST in governmental hospitals at district and sub-district levels were initiated only in 2012 [13]. The adequate coverage of the PWID population, consistent supply of OST drugs, care, support and treatment and retention of PWID in treatment may not have been sustained during the 2012–2016 time period. The effect of OST on HIV prevalence during this time period is beyond the scope of this study and requires further study.

We found that the proportion of HIV among PWID during the 2017–2021 time period was higher than the proportion reported in 2007–2011, and this could be ascribed to the increased access to and use of mobile phones coupled with the greater availability of drugs [26]. This would make it easier to purchase drugs and arrange their delivery at home. A newspaper article published in 2010 revealed that more than half of Mizoram’s population were users of mobile phones [27]. In 2016, mobile phone usage increased in the country with the launch of free 4G internet by Jio [28]. The rise in mobile phone use, led by the availability of cheap Android smartphones, enabled more immediate and convenient ways to arrange drug transactions in Mizoram. Using a dataset from the Drug Use Monitoring in Australia (DUMA) program, a study in Australia found that the increased use of mobile phones has promoted a more immediate and convenient way of transacting drugs [29]. A study that examined face-to-face, in-depth interviews with 21 participants with active drug dealers in Denmark found that using mobile phones for phone calls, texting and messaging apps is a common way to buy, sell and deliver drugs [30]. The increasing use of mobile phones has enabled a more flexible mode of drug distribution, which is less dependent on the physical place [30]. The impact of mobile phones on drug use patterns and HIV among PWID needs further research. However, the proportion of HIV among PWID during the 2017–2021 time period was lower than the 2012–2016 time period, and this may be due to the impact of OST intervention and the introduction of the ‘test and treat strategy’ intervention in 2017 in Mizoram. Under this strategy, people living with HIV were given free ART irrespective of their CD4+ count [31]. However, it is important to note that this study did not evaluate these interventions but intended to provide some direction to the government of Mizoram on the subpopulation to target in future interventions.

We found that female PWID were almost three times more likely to report HIV infection. This finding is supported by a cross-sectional survey conducted in Tanzania, which reported that female PWID were vulnerable to HIV infection due to multiple risk factors such as engaging in unsafe sexual practices, having multiple sexual partners, engaging in commercial sex work [19] and being physically assaulted and raped by their partners [32,33]. Strategies to reach and engage women in HIV prevention interventions are urgently needed.

The findings of our study also suggested that being divorced/separated/widowed was a factor associated with higher HIV among PWID, which is similar to the findings of a study from South Africa [34]. Mizoram has India’s highest divorce rate (6.34%) [35]. A study in Mizoram [35] found infidelity to be the primary cause of divorce, followed by the use of intoxicants. Divorced/separated/widowed PWID have more sexual partners due to a wider sexual network, increasing their risk of HIV/AIDS [36].

Our study also found that having monthly income was positively associated with HIV infection among PWID. This finding is in line with a study from China [37], which found that having a temporary income was an influencing factor for injecting drugs and that generating any amount of income is likely to have supported drug use. A study from Canada [38] found that lower and higher total monthly income among PWID was linked to high-risk income generation strategies as well as a range of drug use patterns and a higher intensity of drug addiction and HIV risk.

Our study found that employed PWID had higher odds of HIV infection. Previous studies [39,40] have shown that a substantial proportion of PWID generates income from prohibited activities, and those that engage in such activities possess higher-intensity addiction [41]. Prohibited sources of income include illegal dealing and sex trade work [42]. PWID with high-intensity addiction engaged in prohibited income-generating behaviour to finance drug use [43]. High-intensity, drug-addicted PWID subsequently has higher exposure to risk factors for HIV infection. Many active PWID have difficulty finding legitimate paid work due to unstable housing, limited employable skills and low levels of education [42]. A community-based study conducted during 2017 in Mizoram on the drug use pattern among PWID, found that the majority of the respondents (81.6%) were unemployed and that selling drugs or sex was their main source of income [44].

This study showed that sharing needles/syringes is still a common practice among PWID despite the efforts made by the MSACS to use needle/syringe exchange services and promote harm reduction services. Lack of access to clean needles/syringes, inconsistent supply of needles/syringes from intervention projects [45,46], criminalization and harassment from police and anti-drug groups [47] contribute to PWID participating in unsafe injecting behaviours. In contrast to our findings, studies from San Francisco [48] and Sweden [49] found a decrease in injection risk behaviours among PWID, which was attributed, but not exclusively, to combined oral substitution therapy (OST), needle exchange programmes and increased support for harm reduction education and access to sterile needles.

Public health responses are needed to address the increases in HIV prevalence among PWID. HIV treatment programs that support treatment linkage and adherence among PWID should be implemented within Mizoram. Increased efforts to test PWID for HIV and linking positive cases to care are vital to reducing new infections. Additionally, it is recommended that the local and central governments impose heavier security at the borders with Myanmar, where heroin and illicit drugs may have seeped through for many years [50]. The churches in Mizoram hold a great deal of influence over the lives of people, and to a great extent, clearly control and redefine the society’s values, norms and morality [51]. They may play an important role in addressing HIV infections in Mizoram. It is recommended that local churches focus on promoting HIV/AIDS awareness and education campaigns, as this would be beneficial for the prevention of HIV in Mizoram [52].

The strength of this study lies in the fact that this is the first study to analyse the factors associated with HIV/AIDS burden among PWID in Mizoram. The second strength is that this is the first population-based study with a large sample size that better represents the PWID population in Mizoram. Because of this large representation, the evidence is potentially more beneficial to guide policy interventions related to HIV prevention. However, there are also limitations to this study. First, there is a potential for selection bias as the participants who were only willing to participate in TI services were registered and included in MSACS data. Secondly, our estimates are potentially limited as they were created using secondary data that were available to us, and this may have contributed to either an underestimation or overestimation of HIV prevalence, thus limiting our ability to interpret trends in HIV prevalence among PWID accurately. Thirdly, desirability bias is possible, with respondents giving socially desirable responses rather than the truth [53].

## 6. Conclusions

Over a period of 15 years, the trend in the prevalence of HIV among PWID remains high in Mizoram. HIV among PWID was significantly higher among female and divorced/separated/widowed participants and among PWID who had a monthly income. A large proportion of PWID still engages in sharing needles/syringes. Continued scaling up of harm reduction services, including uninterrupted needle/syringe exchange services and ART services and adherence among HIV-positive PWID, could be keys to averting further HIV infection among PWID.

## Figures and Tables

**Figure 1 ijerph-20-05871-f001:**
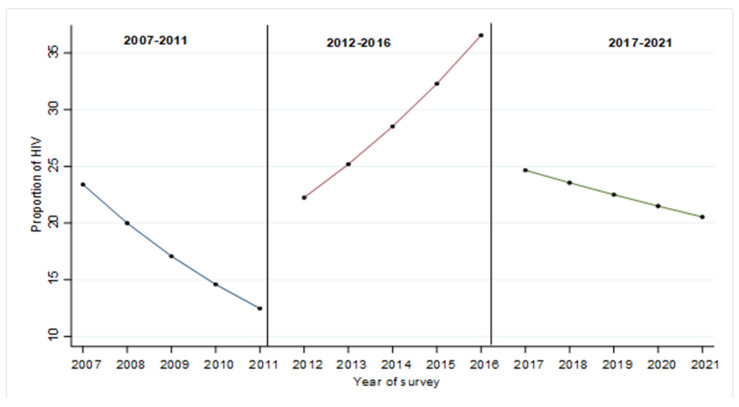
Changes in the proportion of HIV by survey period 2007–2011, 2012–2016 and 2017–2021.

**Table 1 ijerph-20-05871-t001:** Sociodemographic characteristics, injecting and sexual behaviours of PWID for three time periods (2007–2011), (2012–2016) and (2017–2021).

Characteristics	Years 2007–2011	Years 2012–2016	Years 2017–2021
	Total *n* (%)	Total *n* (%)	Total *n* (%)
Gender			
Male	4405 (95.16)	3173 (89.41)	6077 (92.01)
Female	224 (4.84)	376 (10.59)	528 (7.99)
Age			
18–24	2522 (55.20)	1691 (48.49)	2522 (39.36)
25–34	1912 (41.85)	1546 (44.34)	2974 (46.41)
>35	135 (2.95)	250 (7.71)	912 (14.23)
Marital status			
Never married	2736 (59.41)	2250 (63.76)	3986 (60.41)
Married	1100 (23.91)	667 (18.90)	1498 (22.70)
Separated/divorced/widowed	765 (16.63)	612 (17.34)	1114 (16.88)
Education status			
Primary (0–6 years)	924 (20.03)	501 (14.12)	572 (8.69)
Middle (7–9 years)	1810 (39.24)	1262 (35.46)	1960 (29.78)
Higher (10–12 years)	1824 (39.54)	1678 (47.28)	3662 (55.65)
Graduate and above	55 (1.19)	108 (3.04)	387 (5.88)
Employment status			
Unemployed	2530 (54.85)	1800 (50.75)	3337 (50.53)
Employed	1766 (38.28)	1436 (40.46)	2233 (33.81)
Self-employed	317 (6.87)	313 (8.82)	1034 (15.65)
Average monthly income (INR)			
None	1978 (42.95)	1311 (37.35)	2028 (30.93)
<3000	1093 (23.74)	1234 (35.16)	2083 (31.77)
3001–6000	1103 (23.95)	612 (17.44)	1435 (21.89)
6001–10,000	357 (7.75)	238 (6.78)	608 (9.27)
>10,000	74 (1.61)	115 (3.28)	402 (6.13)
Sharing of needles/syringes			
No	3910 (85.02)	3008 (85.12)	3638 (55.51)
Yes	689 (14.98)	526 (14.88)	2916 (44.49)
Condom use with regular partner			
No	1124 (25.28)	1099 (35.54)	2327 (38.85)
Yes	3322 (74.72)	1993 (64.46)	3662 (61.15)

**Table 2 ijerph-20-05871-t002:** Changes in the prevalence of HIV infection among PWID and significance of changes in sociodemographic, injecting and sexual behaviour.

Characteristics	2007–2011 HIV-Positive Prevalence % (95% CI)	2012–2016 HIV-Positive Prevalence % (95% CI)	2017–2020 HIV-Positive Prevalence % (95% CI)	2007–2011 and 2012–2016	2012–2016 and 2017–2021	2007–2011 and 2017–2021
				% (95% CI)	% (95% CI)	% (95% CI)
	N = 4629	N = 3549	N = 6605			
Gender						
Male	13.1 (12.1–14.1)	26.9(25.4 -28.5)	20.8 (19.8–21.8)	13.8 (11.9 to 15.6) ***	−6.0 (−7.0 to −4.2) ***	7.7 (6.2 to 9.1) ***
Female	18.3 (13.7–23.9)	53.9 (48.8–58.9)	41.7 (37.6–46.0)	35.6 (28.4 to 42.8) ***	−12.1 (−18.7 to −5.5) ***	23.4 (16.8 to 30.1) ***
Age						
18–24	12.8 (11.5–14.1)	29 (26.9–31.3)	21.1 (19.6–22.8)	16.2 (13.7 to 18.8) ***	−7.9 (−10.6 to −5.2) ***	8.3 (6.2 to 10.4) ***
25–34	14.6 (13.1–16.3)	31.1 (28.8–33.5)	23.0 (21.5–24.5)	16.5 (13.6 to 19.3) ***	−8.1 (−10.8 to −5.3) ***	8.4 (6.1 to 10.6) ***
>35	8.1 (4.6–14.1)	25.5 (20.5–31.3)	24.1 (21.5–27.0)	17.3 (10.2 to 24.5) ***	−1.4 (−7.4 to 4.7) ^NS^	16 (10.6 to 21.4) ***
Marital status						
Never married	12.4 (11.2–13.7)	27.0 (25.2–28.9)	19.3 (18.1–20.5)	14.7 (12.4 to 16.9) ***	−7.6 (−9.9 to −5.4) ***	6.9 (5.1 to 8.6) ***
Married	11.8 (10.0–13.9)	26.0 (22.8–29.5)	21.6 (19.6–23.8)	14.2 (10.4 to 18.5) ***	−4.4 (−8.3 to −4.8) *	9.7 (6.9 to 12.6) ***
Separated/divorced/widowed	18.3 (15.7–21.2)	43.9 (40.0–47.9)	35.0 (32.2–37.8)	25.6 (20.8 to 30.4) ***	−8.9 (−13.8 to −4.1) ***	16.7 (12.7 to 20.6) ***
Education status						
Primary (0–6 years)	9.5 (7.8–11.6)	25.0 (21.4–29.0)	24.8 (21.5–28.5)	15.5 (11.2 to 19.8) ***	−0.2(−5.37 to −5.1) ^NS^	15.3 (11.3 to 19.3) ***
Middle (7–9 years)	13.9 (12.4–15.6)	31.6 (29.1–34.3)	23.4 (21.6–25.3)	17.7 (14.7 to 20.7) ***	−8.2 (−11.3 to −5.0) ***	9.4 (7.0 to 11.9) ***
Higher (10–12 years)	14.7 (13.1–16.4)	29.9 (27.8–32.2)	21.8 (20.5–23.2)	15.2 (12.5 to 17.9) ***	−8.1 (−10.8 to −5.3) ***	7.1 (5.0 to 9.2) ***
Graduate and above	12.7 (6.2–24.4)	27.1 (19.5–36.3)	21.2 (17.4–25.5)	14.4 (2.2 to 26.6) *	−5.9(−15.2 to 3.4) ^NS^	8.4 (−1.2 to 18.2) ^NS^
Employment status						
Unemployed	11.6 (10.4–12.9)	28.9 (26.9–31.1)	18.9-(17.6–20.3)	17.3 (14.9 to 19.8) ***	−10.0 (−12.5 to 7.5) ***	7.3 (5.4 to 9.1) ***
Employed	16.5 (14.9–18.4)	31.3 (28.9–33.7)	27.7 (25.9–29.6)	14.7 (11.7 to 17.7) ***	−3.5 (−6.6 to −0.7) *	11.2 (8.6–13.7) ***
Self-employed	9.2 (6.5–12.9)	27.4 (22.7–32.7)	22.6 (20.1–25.2)	18.2 (12.3 to 24.1) ***	−4.8 (−10.4 to −0.7) ^NS^	13.4 (9.3–17.4) ***
Average monthly income (INR)						
None	11.9 (10.6–13.4)	23.3 (21.1–25.7)	16.7 (15.1–18.4)	11.3 (8.7 to 14.1) ***	−6.6 (−9.4 to −3.8) ***	4.7 (2.5 to 6.9) ***
<3000	17.3 (15.2–19.7)	35.5 (32.9–38.3)	27.5 (25.6–29.4)	18.2 (14.7 to 21.7) ***	−8.1 (−11.3 to −4.7) ***	10.1 (7.2 to 13.1) ***
3001–6000	11.6 (9.8–13.6)	30.3 (26.8–34.1)	22.3 (20.2–24.6)	18.7 (14.6 to 22.9) ***	−8.0 (−12.3 to −3.7) ***	10.7 (7.9 to 13.6) ***
6001–10,000	12.4 (9.3–16.2)	30.7 (25.1–36.8)	25.3 (22.0–29.0)	18.3 (11.5 to 25.1) ***	−5.3 (−12.1 to −1.5) ^NS^	12.9 (8.1 to 17.8) ***
>10,000	22.9 (14.5–34.1)	29.4 (21.6–38.6)	20.2 (16.6–24.4)	6.5 (−6.5 to 20.1) ^NS^	−9.1 (−18.6 to −0.2) ^NS^	−2.6 (−13.2 to 7.9) ^NS^
Sharing of needles/syringes						
No	11.5 (10.6–12.6)	28 (26.4–29.6)	20.3 (19.0–21.7)	16.4 (14.5 to 18.3) ***	−7.6 (−9.7 to −5.5) ***	8.7 (7.1 to 10.4) ***
Yes	23.1 (20.1–26.4)	39.4 (35.3–43.6)	25.4 (23.9–27.0)	16.2 (11.0 to 21.5) ***	−13.9 (−18.4 to −9.4) ***	2.3 (−1.2 to 5.8) ^NS^
Condom use with regular partner						
No	24.1 (21.7–26.7)	32.6 (29.9–35.4)	22.4 (20.8–24.2)	8.5 (4.7 to 12.2) ***	−10.1 (−13.4 to −6.9) ***	−1.6 (−4.6 to 1.4) ^NS^
Yes	8.7 (7.8–9.7)	26.6 (24.7–28.6)	21.0 (19.7–22.3)	17.9 (15.7 to 20.1) ***	−5.6 (−8.0 to −3.3) ***	12.2 (10.6 to1 3.9) ***

*** *p* < 0.001; * *p* < 0.05; ^NS^ = Non-Significant.

**Table 3 ijerph-20-05871-t003:** Adjusted and unadjusted odds ratios for factors associated with HIV among PWID (the years 2007–2021).

Characteristics	OR (95% CI)	*p*-Value	AOR (95% CI)	*p*-Value
HIV status (*n* = 14,681)				
2007–2011	1		1	
2012–2016	2.75 (2.46–3.07)	<0.001	2.35 (2.07–2.66)	<0.001
2017–2021	1.88 (1.69–2.08)	<0.001	1.41 (1.24–1.59)	<0.001
Gender (*n* = 14,680)				
Male	1		1	
Female	2.85 (2.51–3.23)	<0.001	2.35 (1.81–2.44)	<0.001
Age (*n* = 14,364)				
18–24	1		1	
25–34	1.16 (1.06–1.25)	0.001	1.03 (0.93–1.13)	0.534
>35	1.28 (1.02–1.35)	0.025	0.91 (0.77–1.08)	0.299
Marital status (*n* = 14,639)				
Never married	1		1	
Married	1.01 (0.90–1.11)	0.892	1.13 (1.00–1.27)	0.045
Separated/divorced/widowed	1.99 (1.80–2.20)	<0.001	1.74 (1.54–1.96)	<0.001
Education status (*n* = 14,655)				
Primary (0–6 years)	1		1	
Middle (7–9 years)	1.30 (1.14–1.49)	<0.001	1.24 (1.06–1.44)	0.005
Higher (10–12 years)	1.29 (1.14–1.47)	<0.001	1.13 (0.98–1.32)	0.081
Graduate and above	1.26 (1.00–1.59)	0.048	1.03 (0.79–1.35)	0.798
Employment status (14,677)				
Unemployed	1		1	
Employed	1.43 (1.32–1.56)	<0.001	1.14 (1.03–1.27)	0.011
Self-employed	1.13 (0.99–1.29)	0.052	0.96 (0.82–1.12)	0.649
Average monthly income (INR) (*n* = 14,585)				
None	1		1	
<3000	1.88 (1.70–2.07)	<0.001	2.15 (1.91–2.42)	<0.001
3001–6000	1.27 (1.13–1.42)	<0.001	1.42 (1.23–1.64)	<0.001
6001–10,000	1.46 (1.25–1.71)	<0.001	1.56 (1.29–1.89)	<0.001
>10,000	1.44 (1.17–1.76)	0.001	1.43 (1.11–1.83)	0.004
Sharing of needles/syringes (*n* = 14,610)				
No	1		1	
Yes	1.53 (1.41–1.67)	<0.001	1.78 (1.61–1.98)	<0.001
Condom use with a regular partner (*n* = 13,448)				
No	1		1	
Yes	0.63 (0.58–0.69)	<0.001	0.77 (0.70–0.85)	<0.001

## Data Availability

This study was based on an analysis of the existing dataset from the MSACS. Datasets used in this study are available upon request from the corresponding author.

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
