# Peer review of "Changes in and Predictors of HIV among People Who Inject Drugs in Mizoram, Northeast India, from 2007 to 2021"

_ijerph, 2023, doi:10.3390/ijerph20105871_

Round 1

Reviewer 1 Report

This paper describes the trend of HIV prevalence among PWID enrolled in the targeted intervention programme in an Indian state, and assess the determinants of HIV infection across 3 time points.

Major comments:

·         It is less than intuitive to the reason why chi-square test was used to compare the differences in prevalence, as well as interpreting the odds ratios as such (lines 17-18; 204-205; 221-225).  Additional references would be needed to justify the use of chi-squared test for data collected at two time points (normally it is used to test for two variables within the same population). Would repeated-measures ANOVA be a more appropriate test? A multilevel model could also be more appropriate. If the authors are comparing the odds ratio, then risk ratio could be a more appropriate measure.

·         As the study did not aim to compare the prevalence of HIV among PWID in Mizoram and the other states, and such data was not presented, the discussions are not founded (lines 28).

·         The prevalence of HIV in the three periods was so distinctive that some explanations are warranted, especially at the breakpoints between 2011 and 2012, and between 2016 and 2017. Were there methodological variations? Were the services provided changed across time points to explain the sudden increase of needle-sharing in 2017-2021?

·         Lines 136-139 require a reference.

·         Lines 165-168 are not entirely correct.

·         Lines 168-169 lack a statistical test if the authors claim it ‘increased significantly’.

·         Was the registration of PWID done annually?

·         Confidence intervals should be added to Fig 1.

·         Lines 245-248 require a reference. References 19-22 are too farfetched.

·         Lines 167-273: whether intoxication was the cause or the consequence of the divorce? (Reference 28 is not necessary; even if so it is not an appropriate reference to give a definition).

·         Lines 285-287 require a reference. 

·         Lines 289-290 require a reference. Even if it is true, the authors’ claim are not supported by the data and should be avoided in this paper.

·         Limitations: can the results be generalised to the entire PWID population in the area, if not the title of the paper needs to be changed.

Minor comments:

·         Grammatical errors need to be checked throughout (e.g. in Abstract: “A sample of 14783 PWID participants 14 was extracted from the Targeted Intervention (TI) services under Mizoram State AIDS Control Society (MSACS) for the period 2007-2021were analysed.”, in Results: “Those who earn 211 <3000 INR (AOR=2.14, 95%CI 1.91-2.42), 3001-6000 INR (AOR= 1.42, 95%CI 1.23-1.64), 212 6001-10,000 INR (AOR=1.56, 95% CI 1.29-1.89), >10,000 INR (AOR= 1.43, 95%CI 1.11=1.83).”)

·         Lines 154-156 are not necessary.

·         The adjusted odds ratio could be inverted on Lines 26-27.

Reviewer 2 Report

The study focuses on changes and predictors of HIV among people who inject drugs 2 in Mizoram, India. Using secondary data from targeted intervention programme under the Mizoram State AIDS Control Society, the study enrolled 14783 PWID participants to analyze changes and predictors of HIV among PWID. Results showed a significant decrease in HIV proportion among PWID during 2007-2011, followed by an increase from 2012-2016 and a decrease from 2017-2021. This study also reported that being female, married and divorced/separated/widowed, having a monthly income, employed, and sharing needles/syringes resulted in significantly higher odds of HIV among PWID.

Major comments

·       Could the authors state what the study design is? Analysis of secondary data is not a study design.

·       The sharing of needle/ syringe was low in the years 2007-2011 and 2012-2016 but was higher in the years 2017-2021. If injection drug use is the major route of transmission, why is the prevalence of HIV higher in 2012-2016 when the injection drug use was lower? Could the authors elaborate on this?

·       In the sample, is it possible to know how many of the PWID were also sex workers? It is plausible that this high risk group is driving the prevalence of HIV.

·       Discussion: In the discussion, the statement “We found that the high proportion of HIV among PWID during 2017-2021 time period was higher than the proportions reported in 2017-2021”. This is confusing, it is the same time period that the authors are comparing.

·       The authors have mentioned that mobile phone use could be a contributing factor to the increased prevalence of HIV. However, mobile phone use has progressively increased over time with more mobile phone users now than in 2016, which makes this explanation implausible (as the prevalence of HIV was lower now in the study). Also, the authors did not examine this association in their study.

·       Since the data time period also overlaps with the COVID-19 pandemic, is the low HIV prevalence noted in 2017-2021 related to under-reporting during the COVID-19 pandemic?

Round 2

Reviewer 2 Report

The authors have addressed most of my concerns. I have no further comments.

Author Response

Reviewer 2 does not have additional questions that we need to respond to. Thank you.